# Indigenous language learning in higher education in Ghana: Exploring students' behavioral intentions using an extended theory of planned behavior

**Ernest Nyamekye** ⓘ *

Department of Arts Education, University of Cape Coast, Cape Coast, Ghana

* ernest.nyamekye@ucc.edu.gh

## Abstract

This study investigated the predictors of indigenous language learning from an empirical and theoretical perspective. A quantitative approach was employed to explore the issue using an extended version of Ajzen's Theory of Planned Behavior that incorporates linguistic insecurity in measuring students' language learning intentions. A total of 180 first-year undergraduate students in the Department of Arts Education and the Department of Ghanaian Languages and Linguistics participated in the study. Using Partial Least Squares Structural Equation Modelling (PLS-SEM), measurement and structural models were evaluated. The results indicated that students' attitudes (behavioral beliefs), subjective norms, language learning efficacy, and students' sense of linguistic insecurity significantly predicted their intentions to study indigenous languages. The exogenous variables accounted for 47.8% of the variance in students' intentions to study indigenous languages in higher education. The strongest predictor of intention was subjective norms ($\beta = 0.399$; $p<0.01$), followed by students' sense of linguistic insecurity ($\beta = -0.254$; $p<0.01$), perceived language learning efficacy ($\beta = 0.169$; $p = 0.013$), and language attitude ($\beta = 0.144$; $p = 0.045$). These results underscore the need for concerned stakeholders to foster positive attitudes and address linguistic insecurity to enhance learners' development of positive behavioral intentions toward indigenous language learning.

## Introduction

The impact of globalization, technological advancement, economic pressures, and the influence of the majority and the so-called global languages continue to pose a serious threat to the survival of indigenous languages across the world [1–3]. The world's indigenous languages are, indeed, under the attack of prestigious languages, especially the English language. Despite several efforts and language policies, some scholars constantly argue that there is a possibility that the world will eventually become monolingual [4]. This assumption seems to reflect reality,

**Data Availability Statement:** The data for this study has been deposited in Mendeley data. The data can be accessed through the link below https://data.mendeley.com/datasets/rzjm7gcjxp/1.

**Funding:** The author(s) received no specific funding for this work.

**Competing interests:** The authors have declared that no competing interests exist

especially in Africa, where almost every country or state has developed a strong attachment to the use of prestigious foreign languages either as official or national languages [5–8].

The language issue in Africa has been quite paradoxical. The African continent is, arguably, one of the continents that has paid considerable attention to language maintenance policies. I argue as such because the majority of African countries have devised language policies as well as language in education policies (LEP) that seek to ensure the maintenance of various indigenous languages. Serious attempts have been made to ensure that indigenous languages are used as the medium of instruction—especially in the early grades of education—and also taught as subjects of study [9,10]. Scholars have and continue to affirm that using the child's indigenous language is the most effective means of educating the African child [11–13]. Despite these scholarly affirmations and years of advocacy and awareness creation about the need to safeguard African languages from extinction through education, there seems to be limited progress in revitalizing and reviving endangered African languages as the current language in educational policies in most African countries is not properly implemented [14,15].

There have been serious debates on whether or not the indigenous languages of Africa can develop to the extent of being used as the national language and as the language of education. These debates are always centered on several factors that are considered impediments to the development of indigenous languages. The most cited challenge with the promotion of African languages is multilingualism. Some scholars argue that the limited role of African languages in education is due to the inability of language policymakers to decide which particular language to adopt. Literature suggests that Africa has over 2000 languages. Nigeria alone is said to have 400 languages [3], while Ghana has over 79 languages out of these languages [16]. This difficulty, according to Prah [17], is the reason many assume that it is too late for African languages to be developed and the reason there is a need for Africa to carry on with the language of their colonial masters. Salawu (ibid.) points out, however, that the multiplicity of languages in Africa may not be the real problem; rather, it is the criteria adopted by linguists in counting the number of languages present in a given country. Using Nigeria as a case, Egbokhare and Oyetade [18] write:

> Besides, if we take the population of those who speak the 10 major and medium [sic] languages either as first or second languages in Nigeria, we would cover close to 90% of the population. There is no reason for instance why Efik, Ibibio, and Anang should be listed as different or why a large number of the Edoid languages are listed as separate autonomous languages.

Their argument seems to challenge the idea that multilingualism is a threat to the development of African languages. Though Africa is largely multilingual, most of these languages are mutually intelligible. Policymakers would require less effort if they attempted to harmonize these languages to aid educational and national development. I, therefore, argue that the most prominent factor that appears to be affecting the development of African languages is *anglo-normativity* [3].—i.e., a situation in which people are coerced to be proficient in English and use it for all communicative activities while considering indigenous languages as inferior and hence marginalized [19,20].

It has been shown that in Africa, major stakeholders—the government, parents, and educated elites—tend to exhibit significantly negative attitudes towards speaking the indigenous languages within school compounds [15,21,22]. Oftentimes, speaking the indigenous languages is considered a disgraceful act, and students are sometimes punished accordingly. While many see this as a condemnable act, Roemer [23] shows that in the Tanzanian context, for instance, teachers have been commended for expediting students' acquisition of English by

using coercion such as punishment (both physical and psychological). As reported by Roemer, some students thank the teachers for using corporal punishment to accelerate their second language acquisition. Such is a shocking revelation and an indication that this education practice has already altered their perspectives on the significance of the first language (L1). This, from my standpoint, has a disastrous implication on the student's view on the relative importance of pursuing indigenous language in school, especially at the tertiary level of education in most African countries, including Ghana. Owu-Ewie and Edu-Buandoh [22], for example, have shown that given the negative attitudes usually attached to the learning of indigenous languages in Ghana, students who are offered indigenous languages as part of the senior high school programmes usually accept it reluctantly. At the university level, there seems to be a dearth of literature that explains, comprehensively, the predictors of native language learning. Hence the need for a study of this nature.

## Justification for the present study

Indigenous languages continue to play a significant role in the development of Ghana as they are used as media of communication in domestic activities, at home, in official programs, in the media, in national addresses, and as languages of instruction in the early stages of childhood education. Given the significant role of indigenous languages in Ghanaian society, they are offered as subjects of study at all levels of education in Ghana. Several higher educational institutions in Ghana continue to admit students to study some indigenous languages in Ghana. Among some of these languages are Akan (Twi and Mfantse), Nzema, Dagaare/Wale, Dagbani, Dangme, Ewe, Ga, Gonja, Gurune, and Kasem. In the University of Cape Coast, for instance, three of these languages—i.e., Akan (Twi and Fante), Ga, and Ewe—are the only indigenous languages offered in the Department of Arts Education and the Department of Ghanaian Languages and Linguistics. Per the academic structure of both departments, first-year students are always given the opportunity to either major, minor, or drop any course as they progress to the second year. Per my observations, the indigenous language courses have, in successive years, recorded a lower number of students. The majority of the students focus on majoring in other courses such as English, history, religion, and human values. This observation provoked my curiosity; we, thus, sought to investigate the determinants of the current batch of first-year students in both departments to gain a better understanding of the situation from a quantitative and a theoretical perspective.

A study of this nature is deemed important because the available literature regarding the factors affecting the study of indigenous languages in education has limited empirical support and is mostly based on mere speculations. For instance, Guerini [24] asserts that students at the university level may feel reluctant to study indigenous languages because those who study indigenous languages are often seen as underachievers who turn to easier options to obtain good grades. Several other scholars have reiterated that the negative attitudes of people toward those who study indigenous languages, especially at the senior high school level, contribute to the low level of student enrollment [22,25,26]. Nonetheless, these speculations are yet to be opened to empirical testing, especially in higher education. The current study, therefore, seeks to investigate these problems empirically using the theory of planned behavior (TPB) to explore possible predictors of first-year students' intention to study indigenous languages in higher education given the scarcity of literature on the topic.

## Theoretical framework: The theory of planned behaviour (TPB)

The current study is rooted in the TPB. I conducted this study within this theoretical framework because I had the suspicion that students' decision to further their undergraduate studies

at the university may depend not only on their willingness to do so but also on external factors such as the stereotypical behavior associated with the study of indigenous languages at all levels of education in Ghana and African countries alike, their perception of the relative difficulty of the language they are studying, and their personal attitude towards the study of that language. I considered the TPB a suitable framework for the current study because the focus of the current study aligns well with the core tenets of the theory. The next section discusses the core tenets of the TPB and presents a theoretical model that illustrates the structural link between how human behavior could be influenced by this salient information. It finally reviews some relevant studies that show how TPB has been applied in the field of language learning.

## TPB explained

As Ajzen [27] puts it, "Explaining human behavior in all its complexity is a difficult task." Behavior, according to this scholar, could be measured by considering various processes: the physiological process at one end and the influence of social institutions at the other end. Such consideration appears to be the very reason the TPB has emerged as one of the most frequently applied theories for measuring human behavior. It measures human behavior by taking into consideration the complexities involved. TPB explores three major considerations that account for an individual's intention to exhibit or perform a particular behavior: behavioral beliefs, normative beliefs, and control beliefs [27,28]. The theory assumes that every behavior is dependent on salient information or beliefs (behavioral beliefs, normative beliefs, and control beliefs) about the behavior. These beliefs, according to the TPB, are the formidable determinants of a person's intention and actions in a given situation. Fig 1 explains how the various constructs in Azjen's TPB come together to predict people's intention to perform a particular behavior.

As illustrated in Fig 1, the basic structural assumption of the TPB is that intention is the immediate antecedent to behavior [27,29,30]. Nonetheless, an individual's intention to perform a behavior is influenced by their attitudes towards the behavior, subjective norms, as well as the individual's perceived sense of efficacy toward the behavior [31,32]. Perceived control beliefs have both direct and indirect effects on the performance of a behavior. The indirect effect of control beliefs through intention on behavior is based on the assumption that individuals who are efficacious in accomplishing a particular task may have positive intentions to perform a particular task or behavior. In other words, the more individuals think they possess the requisite capabilities, resources, and opportunities, the greater their perceived control over

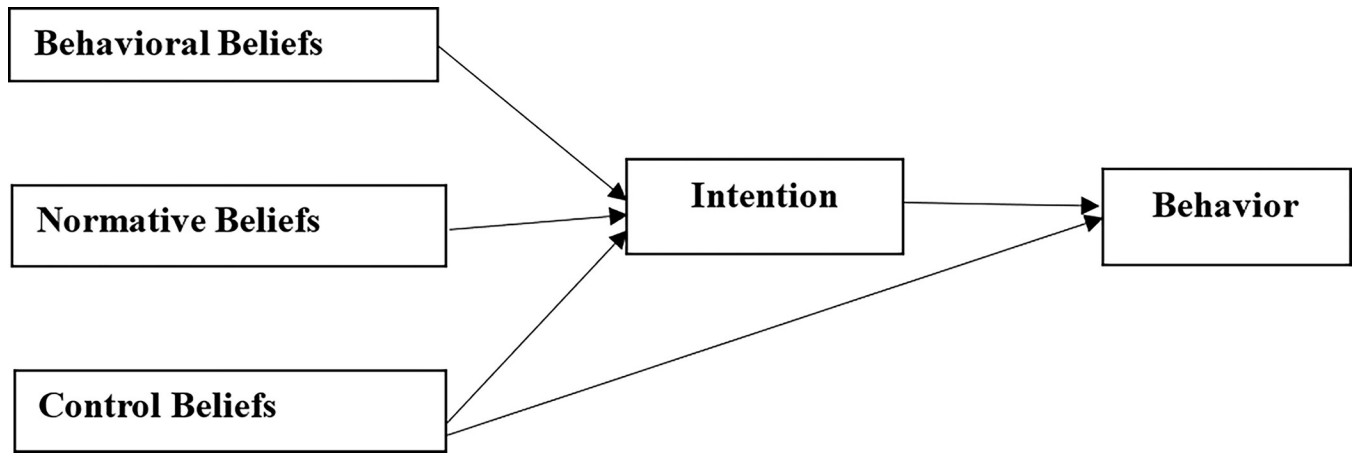

**Fig 1. Azjen's theoretical model.**

their behavior [29]. An individual's self-efficacy beliefs (their perceived control over the behavior) are, thus, considered the only construct that could have a direct effect on the performance of the behavior.

## Conceptualization and development of hypotheses

Under this section of the study, I conceptualize the proposed theoretical model which grounds the conduct of this study. Based on previous literature, I dedicate this section to the discussion of the theoretical influence of the TPB predictors (attitude, perceived behavioral control, subjective norms) on learners' behavioral intention to learn indigenous languages. The novelty of the proposed model is that I incorporate and discuss how linguistic insecurity is theoretically understood as a predictor of individuals' intention to learn language.

*Attitude (ATT)*: Attitude toward a behavior has been viewed as one of the determinants of a particular behavior. It is conceptualized as an individual's evaluation of a particular behavior of interest. In line with the present study, the behavioral beliefs would relate to the students' judgment of the relative importance of learning indigenous languages to their lives. As Alhamami [33] posits, student attitude toward a behavior is dependent on whether the behavior is negatively or positively valued. Hence, I hypothesized as follows:

H1: *attitude will influence students' intention to study indigenous languages.*

*Linguistic Insecurity (LI)*: In this study, linguistic insecurity refers to speakers' hesitation to speak or learn their native language due to negative perceptions of them. Though the research on this issue is rather limited in the Ghanaian setting, previous studies have indicated students prefer to change to the English language when they desire to present a good self-identity for themselves [34]. This shows that, in some cases, using the local language may be avoided since it does not reflect one's intellectual identity. Linguistic insecurity is associated with language shifts in multilingual settings, according to research. The more individuals feel ashamed of using a less dominant language, the more likely they are to switch to a dominant language with worldwide or state-wide recognition. I therefore hypothesize that students' sense of linguistic insecurity could influence their willingness to learn their indigenous language. Hence, I test the following hypotheses:

H2: *Students' sense of linguistic insecurity will influence their behavioural intention to study indigenous languages at the university level*

*Language learning Efficacy (LLE)*: The apparent ease or difficulty of acting is known as perceived behavioral control. This construct includes control beliefs, which are views on the existence of elements that might help or hinder the performance of the behaviors [27]. I apply this construct in the context of this study to measure the extent to which students believe they have the requisite potential and strength to learn Indigenous languages at the university and how it influences their attitudes and intention to pursue local language-related courses at the university. Finally, I hypothesize that students' intention to take further indigenous courses could stem from their perceived behavioral control.

H3: *Students' language learning efficacy will influence their intention towards indigenous language learning.*

*Subjective Norms (SN)*: Another predictor of an individual's intention to perform a given behavior is the normative expectations of others (subjective norms). This is also viewed as an influential factor in people's actions. Subjective norms, as a construct in the TPB, take into consideration how social pressures account for an individual's intention to perform a

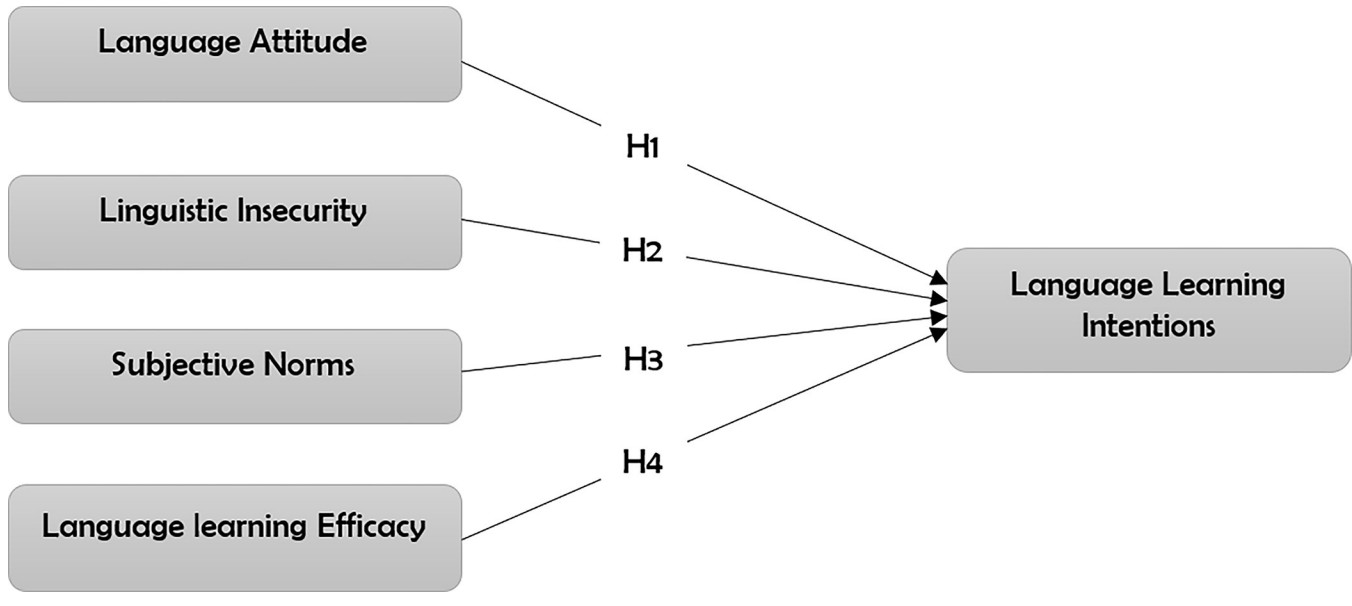

**Fig 2. Proposed conceptual model.**

particular behavior [29]. The construct is applicable in the current study because I hypothesize that students' intentions to take further indigenous language courses at the university could also be a function of what people they value say about the need for pursuing such courses. In the context of this study, I also assume that what students hear about the study of indigenous languages (subjective norms) in school could influence their language-learning intentions. On this basis, I hypothesize as follows:

H4: *subjective norms will influence students' intention to study indigenous languages*.

Fig 2 is a conceptual framework that summarises the various hypotheses set to guide the current study.

## Studies on the application of TPB language learning and use

Literature on the application of TPB to studying indigenous languages, especially in the African context, is quite scarce. A few attempts have, however, been made among researchers in foreign contexts. Zhong [35] studied immigrants' willingness to communicate in L2 classrooms. This particular study was based on the researcher's realization of some learners' reluctance to communicate in the second language and was therefore motivated to explore factors that accounted for such reluctance among the language learners. The researchers explored the predictors of learners' willingness to communicate using multiple qualitative research methods (in-depth interviews, classroom observation, etc.). Drawing on the TPB as a theoretical framework, the researcher discovered that linguistic factors, socio-cultural factors, self-efficacy, and learner beliefs accounted for learners' willingness to communicate in the L2 classroom. Girardelli and Patel [36] also observed Chinese students' reluctance to communicate in American classes using the TPB. The aim was to understand why students may feel notoriously reluctant to communicate in the classroom. The data was gathered from 133 Chinese students enrolled in Sino-American universities. The partial least squares path modeling analysis indicated a 39% variance in students' intention to participate in class. Students' attitudes and self-efficacy beliefs were the strongest predictors of intention. With an extended version of the TPB,

Girardelli and Patel [37] investigated Chinese EFL learners' participation in the classroom. The researcher added two other constructs (foreign language anxiety and face-saving), which were frequently discussed among Chinese EFL learners. Using partial least squares modeling analysis, it was discovered that all the constructs of the extended TPB explained 46% of the variance in the learners' intentions to participate in using the English language in the classroom.

With respect to language learning, a few attempts have also been made by some scholars to understand the predictors of intention to learn. Rahman and Mahmud [38], for instance, conducted a study that sought to explore the L3 learning intentions of students. The study considered 161 international students studying in the Chinese Language Learning program in Yunnan, Beijing, Jiangsu, and Hunan. An extended TPB that included Self-Determination Theory (SDT) and the influence of L2 was used for the analysis. Using the PLS, the study indicated that, apart from subjective norms, all the other constructs of the TPB and SDT, as well as students' L2 proficiency, were significant predictors of students' intention to study L3.

The reviews above indicate how the TPB could be used to provide a comprehensive picture of the determinants of language use and language learning. Nonetheless, the available studies are more focused on foreign languages. In terms of African indigenous language use and learning, the available studies do not seem to provide any empirically and theoretically valid explanations for the determinants of language use and language learning. A significant number of studies across the continent focus on factors militating against the L1 medium of instruction [26,39–42]. In terms of learning the first languages within the classroom context, there seems to be a scarcity of empirical literature indicating some of the determinants of students' intention to study. Scholars [22,25,43] constantly lament that, given the negative attitudes associated with the learning of indigenous languages, students are always reluctant to pursue indigenous languages at various educational levels. The current study therefore seeks to fill the identified empirical gap by adopting the TPB to explore the pressing determinants of Ghanaian university students' intention to learn indigenous languages.

## Research methods

### Study design and participants

A cross-sectional survey design was adopted for this investigation. Out of the 258 first-year undergraduate students in the Department of Arts Education and the Department of Ghanaian Languages and Linguistics at the University of Cape Coast, 180 participated in the study through a simple random sampling technique. Based on the Krejcie and Morgan [44] sampling size threshold, 180 was considered representative enough to support the generalization of findings. These participants were involved in this study because they all undertake compulsory courses in one of the three indigenous languages (Akan, Ewe, and Ga). However, these students have the option to either minor or major in various courses offered in the two departments. As has been observed in previous years, only a few students take the various indigenous languages as their minors. Thus, I considered this batch of students appropriate for this investigation, as they are on the verge of deciding whether or not they would further the study of indigenous languages as they progressed into their second year.

### Instrumentation

Data for this study was gathered with a closed-ended questionnaire based on the TPB. The questionnaire was composed of two main sections. The first section gathered information on the students' demographic background, including their gender, program of study, and the specific indigenous language they are into. The other sections of the questionnaire were based on the constructs in the TPB: attitude, subjective norms, perceived behavioral control, and

intention. The questionnaire excluded the 'behavior' aspect of the TPB because the study did not consider examining their actual language learning behavior. The questionnaire was designed based on a five-point Likert scale.

## Measures

The data gathered for this study was analyzed using the PLS-SEM. SmartPLS 4 statistical software was employed to aid the analysis. This statistical approach was considered for the current study because it is usually deemed an appropriate analytical tool when dealing with complex statistical models involving multiple latent and observed variables [45,46]. Both measurement and structural models were evaluated. In the evaluation of the measurement model, Cronbach's alpha and the composite reliability of the outer model were assessed. Also, the convergent validity of the outer model was assessed through the average variance extracted. Also, the validity of the model was further assessed through discriminant validity assessments. The three major criteria for measuring discriminant validity—i.e., the heterotrait-monotrait ratio (HTMT) criterion and the Fornell-Larcker criterion—were employed to establish how distinct the latent variables are from each other.

## Ethical considerations

Because this study involved humans, the techniques detailed in this section were used to conform to research ethical rules. The researchers initially obtained ethical permission from the University of Cape Coast's institutional review board before collecting data from the sampled students. After obtaining the research ethical approval, questionnaires were administered to the research participants on November 10, 2023, and ended on January 2, 2024. Before collecting data, students were also asked for their permission. To confirm their participation in the survey, they were required to sign a written consent form. Most importantly, participants were informed that the information they provided in the questionnaire would be kept confidential.

## Results

This section presents and analyzes the results of the study. We, foremost, present the demographic data of the respondents involved in the research. Following this are the assessments of the measurement and structural model.

### Demographic data of respondents

Table 1 presents the demographic composition of the participants in the study. The total number of respondents for the study was 180. Of these, male students constituted 41.7% of the total sample, while the remaining 58.3% were females. The percentage of students pursuing a Bachelor of Education in Arts was 45.6%, while those pursuing a Bachelor of Arts were 26.1%. The remaining 28.3% were offering communication studies.

### Measurement model assessment

The measurement model was assessed to ascertain the overall validity and reliability of the constructs. The internal consistency, convergent, and discriminant validity assessments proved that the model was appropriate for further analysis.

**Internal consistency assessment.** As Hair and Risher [47] indicate, it is imperative to check the validity and reliability of the constructs by considering the composite reliability, internal consistency using Cronbach's alpha (α), and the constructs' average variance extracted (AVE). The first assessment of reliability, according to Hair and Hult [48], is the indicator

**Table 1. Respondents' demographic data.**

| Variable | Frequency | Percentage |
|---|---|---|
| **Gender** | | |
| male | 75 | 41.7 |
| female | 105 | 58.3 |
| Total | 180 | 100.0 |
| **Programme of study** | | |
| Bachelor of Education in Arts | 82 | 45.6 |
| Bachelor of Arts | 47 | 26.1 |
| Communication Studies | 51 | 28.3 |
| Total | 180 | 100.0 |
| **Language Studied** | | |
| Fante | 60 | 33.3 |
| Asante Twi | 94 | 52.2 |
| Ga | 6 | 3.3 |
| Ewe | 15 | 8.3 |
| Akuapem Twi | 5 | 2.8 |
| Total | 180 | 100.0 |

Three indigenous languages are offered at the university: Akan, Ga, and Ewe. However, the Akan language has three major dialects that are offered separately: Asante Twi, Akuapem Twi, and Fante. A percentage of 33.3, 52.2, and 2.8 were currently taking Fante, Asante Twi, and Akuapem Twi, respectively. Only 8.3% and 3.3% were studying Ewe and Ga, respectively. As indicated earlier, these indigenous languages are only compulsory in the first year. Students are yet to decide whether to major, minor, or drop these languages before the start of the second year, which makes this batch of students suitable for the current study.

reliability, which measures the extent to which each indicator's variance is explained by its underlying construct. Ideally, the recommended threshold for indicator reliability is 0.7 or above. However, Hair and Hult [48] show that low loadings between 0.4 and 0.7 could be retained if their removal does not affect the other reliability and validity issues. In my model, indicators that loaded lower than 0.6 were removed because retaining these indicators lowered the AVE of some constructs. Another consideration in the measurement of the reflective model was the evaluation of the internal consistency of the items. To this end, both Jöreskog's [49] composite reliability and Cronbach's alpha were employed. As a general rule of thumb, the reliability of all the constructs was above the 0.7 threshold as presented in Fig 3 and Table 2.

**Convergent validity assessment.** This assessment was carried out to evaluate how well the indicators converge to measure one underlying construct. The AVE of all the constructs—i.e., language attitude (0.529), linguistic insecurity (0.607), language learning efficacy (0.569), subjective norms (0.533), and intention (0.760)—indicates how well the observed variables in the model measured the underlying latent variables in the model (see Table 2).

**Discriminant validity assessment.** Having ascertained the convergent validity of the TPB constructs, I further assessed the discriminant validity of the constructs using the Fornell-Larcker criterion and heterotrait-monotrait ratio (HTMT). This was necessary as it helps to determine how the various constructs in the proposed model are distinct from each other. In other words, it was used to ascertain whether or not I was measuring the same underlying construct. Tables 3 and 4 below present the values obtained for each measure of discriminant validity.

The discriminant validity assessment of the HTMT proves how distinct the constructs are from each other. None of the HTMT values were above the recommended threshold. The consensus is that any value above 0.90 indicates a lack of discriminant validity [50,51].

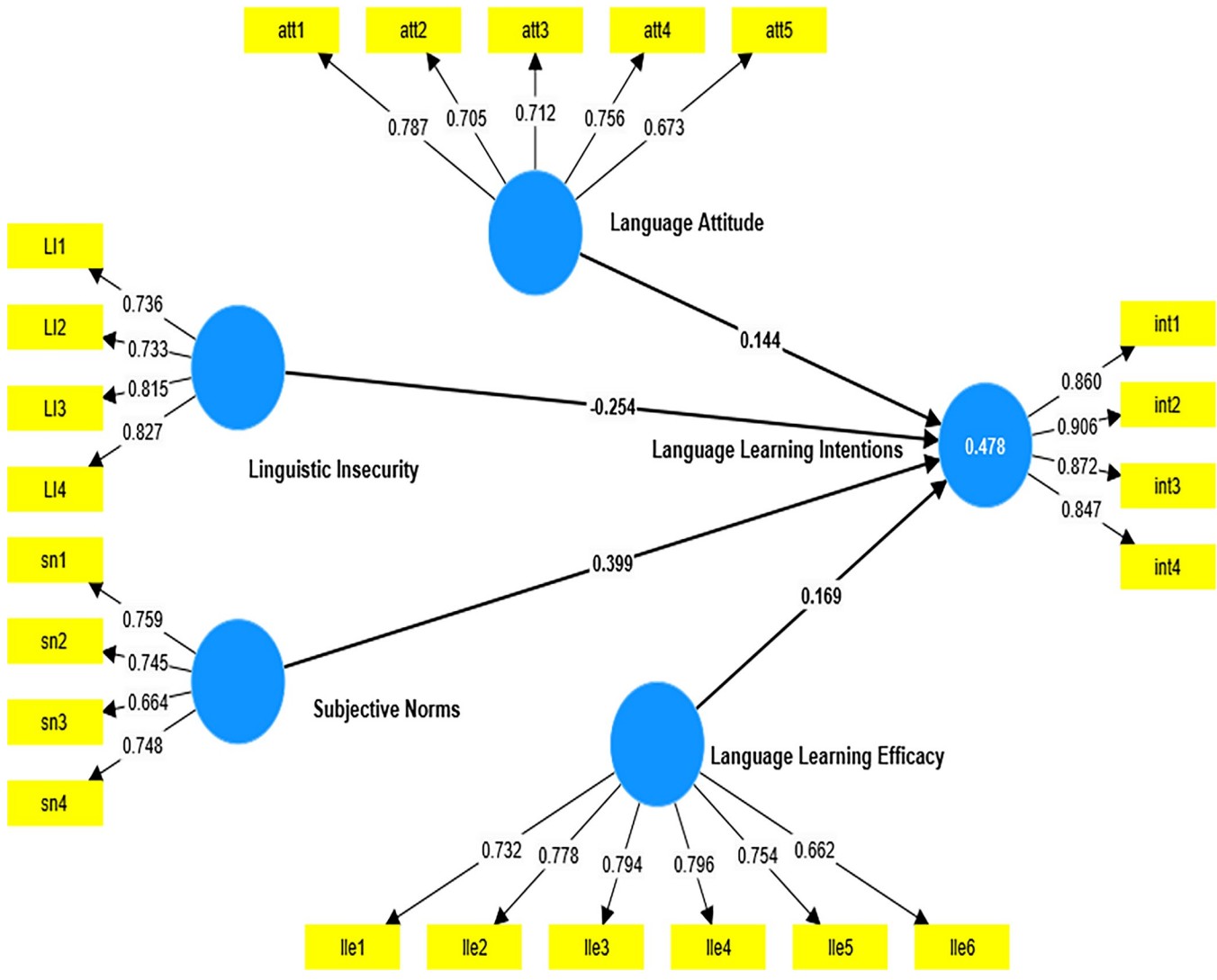

**Fig 3. Measurement model.**

Discriminant validity was also assessed using the Fornell-Larcker criterion, which recommends that the square root of the AVE should be higher than the squared correlation with other constructs [52,53]. As indicated in Table 4, the square roots of all the constructs are higher than the squared correlation with all the other variables, which indicates that the constructs are distinct from each other.

## Multicollinearity assessment

Evaluating the variance inflation factor (VIF) of the model's constructs is considered a necessity. This evaluation is needed to resolve collinearity issues in the model. This evaluation is important because it helps ensure that the predictor variables are not highly correlated with each other. Evaluating and resolving these issues helps the researcher obtain the individual contribution of each predictor variable to the dependent variable [54]. VIF values below 5 are usually considered appropriate since they indicate an absence of collinearity [55]. On this

**Table 2. Outer loadings, Cronbach's alpha, composite reliability and AVE.**

| Construct | Loadings | α | rho a | rho c | AVE |
|---|---|---|---|---|---|
| **Language Attitude (LA)** | | **0.782** | **0.792** | **0.849** | **0.529** |
| att4 | 0.669 | | | | |
| att5 | 0.768 | | | | |
| att7 | 0.703 | | | | |
| att8 | 0.790 | | | | |
| att9 | 0.719 | | | | |
| **Linguistic Insecurity (LIN)** | | **0.785** | **0.805** | **0.860** | **0.607** |
| li1 | 0.736 | | | | |
| li2 | 0.733 | | | | |
| li3 | 0.815 | | | | |
| li4 | 0.827 | | | | |
| **Language learning Efficacy (LLE)** | | **0.847** | **0.848** | **0.887** | **0.569** |
| lle1 | 0.732 | | | | |
| lle2 | 0.778 | | | | |
| lle3 | 0.794 | | | | |
| lle4 | 0.796 | | | | |
| lle5 | 0.754 | | | | |
| lle6 | 0.662 | | | | |
| **Subjective norms (SN)** | | **0.709** | **0.712** | **0.820** | **0.533** |
| sn1 | 0.759 | | | | |
| sn2 | 0.745 | | | | |
| sn3 | 0.664 | | | | |
| sn6 | 0.748 | | | | |
| sn7 | 0.759 | | | | |
| **Language learning intention (LLI)** | | **0.895** | **0.904** | **0.927** | **0.760** |
| Int1 | 0.860 | | | | |
| Int2 | 0.906 | | | | |
| Int3 | 0.872 | | | | |
| Int4 | 0.847 | | | | |

basis, I assume that collinearity issues are not present in my current model, as shown in Table 5 below.

## Common method bias

Collecting data using quantitative instruments like the Likert scale questionnaire usually causes confusion for respondents, especially when these instruments are not well designed. This usually compels respondents to provide consistent answers to different sets of survey

**Table 3. Discriminant validity (HTMT).**

| | LA | LIN | LLE | LLI | SN |
|---|---|---|---|---|---|
| LA | | | | | |
| LIN | 0.298 | | | | |
| LLE | 0.492 | 0.308 | | | |
| LLI | 0.496 | 0.494 | 0.509 | | |
| SN | 0.557 | 0.301 | 0.506 | 0.714 | |

**Table 4. Discriminant validity (Fornell-Larcker criterion).**

|      | LA     | LIN    | LLE    | LLI    | SN     |
|------|--------|--------|--------|--------|--------|
| LA   | **0.728** |        |        |        |        |
| LIN  | -0.244 | **0.779** |        |        |        |
| LLE  | 0.400  | -0.258 | **0.754** |        |        |
| LLI  | 0.433  | -0.423 | 0.451  | **0.872** |        |
| SN   | 0.399  | -0.225 | 0.397  | 0.581  | **0.730** |

questions that are conceptually unrelated. The common method bias, also known as Harman's single-factor test, is a statistical approach that is mostly used to assess spurious statistical estimates, usually caused by undesirable response patterns by the respondents [56]. To ensure the absence of this statistical bias in this study, all the measurement items were factorized into a single factor using the exploratory factor analysis (EFA) technique. After forcing the measurement into a single factor, the result indicated that the first factor accounted for 28.08% of the variance, which is significantly below the cut-off point of 50% [57]. This result indicates a lack of CMB in this study's measurement model.

## Structural model assessment

The hypotheses set to guide the study were assessed in the structural model. To gain a better understanding of the determinants of students' intention to study indigenous languages in higher education, the researcher primarily aimed at examining students' attitudes, their perceived sense of efficacy in studying indigenous languages, and, most importantly, how their social environment (subjective norms) predicted their intention. Fig 4 and Table 6 present the statistical association between the variables in the proposed model.

Table 6 and Fig 4 provide the results of the structural model, which indicate how students' attitude towards their indigenous language, the subjective norms about local language education, students' sense of linguistic insecurity, and language learning efficacy contribute to their intention to learn indigenous languages in higher education. All the hypothesized relationships between the exogenous variables and the endogenous variables were confirmed. Specifically, the results show a positive and marginally significant relationship between students' language attitude and their intention to further their higher education in indigenous language education ($\beta = 0.144$; $t = 2.005$; $p = 0.045$; $f2 = 0.030$). Cohen [58] proposed that f2 values of 0.02, 0.15 or larger, and 0.35 or greater indicate minor, moderate, and large impact sizes, respectively. Hence, the small effect size obtained for this hypothesized relationship suggests that students' language attitudes have a smaller impact on their intention to learn an indigenous language in higher education.

Language learning efficacy was also found to be a significant positive predictor of students' language learning intentions ($\beta = 0.169$; $t = 2.491$; $p = 0.013$; $f2 = 0.042$). This result also

**Table 5. Multicollinearity results.**

| Constructs | LA | LIN | LLE | LLI   | SN |
|------------|----|-----|-----|-------|----|
| LA         |    |     |     | 1.318 |    |
| LIN        |    |     |     | 1.110 |    |
| LLE        |    |     |     | 1.323 |    |
| LLI        |    |     |     |       |    |
| SN         |    |     |     | 1.305 |    |

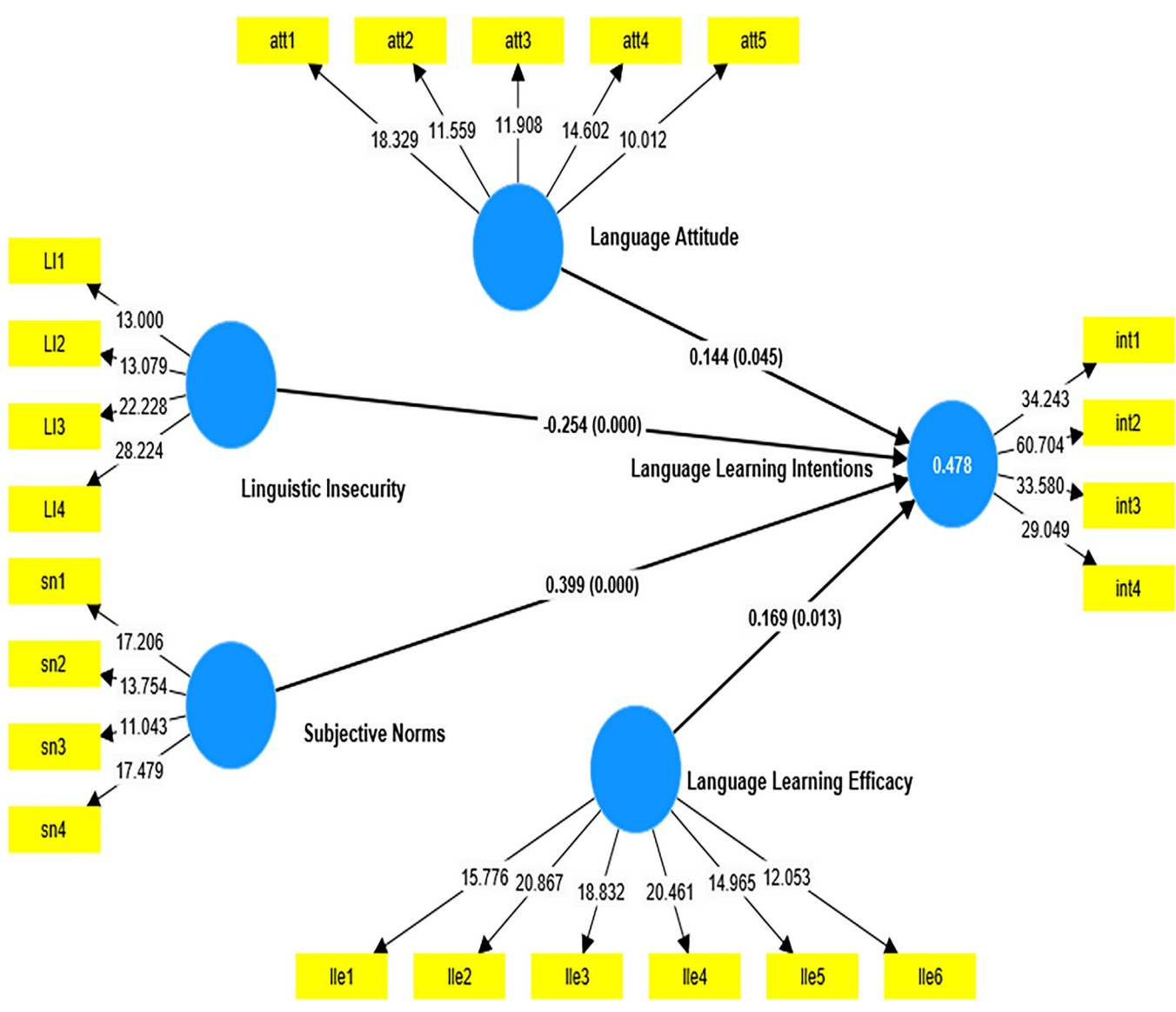

**Fig 4. Structural model.**

indicates that students' sense of confidence regarding their language learning abilities forms part of the host of factors that explain their intention to either further or discontinue the study of indigenous languages at the university. This implies that students with a higher sense of language learning efficacy have higher intentions of learning the indigenous languages, while those with a lower sense of efficacy express a lesser intention. Nonetheless, the effect size

**Table 6. Hypotheses test results.**

| Path | β | SD | t | p | $f^2$ | $R^2$ | $Q^2$ | Decision |
|---|---|---|---|---|---|---|---|---|
| LA -> LLI | 0.144 | 0.072 | 2.005 | 0.045 | 0.030 | 0.478 | 0.439 | Supported |
| LIN -> LLI | -0.254 | 0.061 | 4.181 | <0.01 | 0.111 | | | Supported |
| LLE -> LLI | 0.169 | 0.068 | 2.491 | 0.013 | 0.042 | | | Supported |
| SN -> LLI | 0.399 | 0.061 | 6.581 | <0.01 | 0.234 | | | Supported |

shows that language learning effectively has a lesser impact on students' intentions in this study.

Linguistic insecurity—used in the context of this study to refer to students' sense of shyness and feelings of embarrassment related to pursuing indigenous languages in higher education —was found to be a negative and significant predictor of students' intention to further the study of indigenous languages in the university ($\beta$ = -0.254; t = 4.118; p<0.01; f2 = 0.111). The effect size of this significant prediction was moderate, as suggested by Cohen. It is worth noting, based on the results, that students who feel a higher sense of linguistic insecurity regarding the indigenous languages may be less likely to express an intention to pursue an indigenous language program in higher education.

Finally, the significant positive relationship observed between subjective norms and students' language learning intention is an indication that perceived social pressures and approval from others have a greater influence on the student's intention to learn indigenous languages at the university level ($\beta$ = 0.399; t = 6.581; p = <0.01; f2 = 0.234). The effective size obtained for this hypothesis suggests that social pressures and approvals of significant 'others' have a greater impact on students' intentions compared to all other variables in this current study.

Overall, the results have shown, based on the coefficient of determination ($R^2$), that 47.8% of the variability of students' language learning intentions is accounted for by the four exogenous variables incorporated in this model. It, thus, implies that a significant proportion of the variance 52.2% in students' intentions could be accounted for by exogenous variables not captured in the current model. Additionally, the predictive relevance ($Q^2$) of the model was assessed using the blindfolding procedure. A $Q^2$ value of 0.439 was obtained. According to Hair and Risher [47], predictive relevance is obtained when the $Q^2$ value exceeds 0. However, values that are approximately 0, 0.25, and 0.50 depict small, medium, and large in PLS SEM modeling. Based on these thresholds, it could be said that the current model has substantial predictive relevance, suggesting that the model has been able to make accurate predictions; therefore, accurate generalizations could be made.

## Discussion of results

The study has presented a quantitative assessment of the predictors of first-year undergraduate students' intention to study indigenous languages in the subsequent years. Over the years, only a few students have taken various indigenous languages (Akan, Ewe, and Ga) as their major subjects as they progress to the second year. Though existing studies have indicated that students exhibit an unwillingness to study Ghanaian indigenous languages, there are not enough empirical studies explaining the determinants of such unwillingness. This study has therefore provided a novel insight into the predictors of students' willingness to study indigenous languages at the university level. Using an extended TPB, the study has confirmed that prominent among the factors that have been discussed as impediments to the study of indigenous languages in Ghana are people's perceptions and comments used to describe students who study indigenous languages, especially at the senior high school level. Several scholars [22,59] have argued that studying an indigenous language in Ghanaian schools is seen as a waste of academic careers. According to Owu-Ewie and Edu-Buandoh [22], major stakeholders in education—i.e., parents, teachers, and schools—do not seem to appreciate the importance of learning the local languages in schools. Oftentimes, students who, by any means, study one of the local languages are perceived as underachievers. To some extent, teachers of such indigenous languages are disregarded as far as academic discourses are concerned. It is, thus, quite embarrassing to be identified as a student of an indigenous language. Such conception appears to have a significant effect on students' intentions to study indigenous languages, even at the

university level. As seen in the current study, the subjective norm, which dealt with what people say about studying an indigenous language, appeared as the strongest predictor ($\beta = 0.399$) of students' intention to study indigenous languages.

The study has also confirmed previous assumptions on how students' regard for a language could influence their willingness to use or learn a particular language [60,61]. I have confirmed in this study that, indeed, individual affective states have a significant impact on their language learning intentions. Attitude towards language was seen as a significant contributor to indigenous language learning. This suggests that, possibly, the continual decline in enrolment of students in indigenous language-related courses is somewhat contingent on their negative attitude towards these languages, as scholars have previously suggested [15,22,26]. It has, most importantly, been shown that linguistic insecurity, characterized by students' arousal of negative emotions regarding the study and speaking of indigenous languages in educational terrain in Ghana, could lead to a significant decline in students' intention to further their studies in indigenous language programs. It follows, therefore, that after the mandatory study of indigenous language courses in the first year, students who experience high levels of linguistic insecurity are likely to opt out of indigenous language programs.

Much of the scholarly debate on the topic under review has focused on attitudinal issues [22,62,63]. Nonetheless, studies have confirmed that one of the strongest predictors of intention to learn a language is self-efficacy [35,37,38]. In this study, it has been observed that an important predictor of first-year undergraduate students' intention to continue the study of indigenous languages relates to their perception of the relative difficulty of studying indigenous languages and their perceived ability to do so. The assessment of the structural model has shown that perceived behavioral control is the strongest predictor of students' intentions, as it explains 33.8% of the variance in behavioral intentions. Attitude, on the other hand, did not explain much of the variance in students' intentions to study indigenous languages at the university, though its predictive power was significant.

## Implications of the study

### Theoretical implications

The study has shown the applicability of TPB in explaining comprehensively the determinants of the study of indigenous languages at the university level. Contrary to the previous conceptions, the TPB has helped in understanding that students' willingness to study indigenous languages is not only a function of their attitudes and people's perceptions of the relative importance of these languages. Rather, their perceived control over the study of indigenous languages in educational domains also plays a crucial role. Most importantly, it should be noted that since the causal relationships between the TPB model for this study were all significant, it is assumed that favorable subjective norms, attitudes, linguistic security, and language learning efficacy can positively affect students' intentions to study indigenous languages, while unfavorable subjective norms, attitudes, linguistic insecurity, and efficacy can negatively affect students' intentions to study indigenous languages.

### Managerial implications

Based on the results of the current study, I suggest various recommendations and interventions that could enhance students' intentions to pursue indigenous language courses in higher education. Foremost, the present study's findings underscore the need for policymakers and concerned stakeholders to underscore the significance of adopting a comprehensive policy that considers the social, psychological, and educational factors that effectively promote the intention to learn indigenous languages within the higher education context in Ghana. In

other words, interventions like the national accelerated literacy program, which was initiated to help students develop literacy in their native languages, should meticulously address social norms and perceptions of local language education. This could be made effective if language training programs that aim to equip teachers with skills to establish a supportive and inclusive learning environment for all students are offered to elementary school teachers. This is necessary because training learners to appreciate the cultural and educational benefits of the local languages could mitigate the negative perceptions, which consequently affect students' intentions to pursue language-related courses in the higher education context.

## Limitations and implications for future research

The present study presents some limitations that warrant the conduct of further research to explore the determinants of students' intentions to study indigenous languages. Foremost, the research sample was limited to University of Cape Coast first-year students. The generalizability of the findings to all students in Ghana would not be appropriate. We, therefore, recommend that further studies be conducted to take into consideration students in other tertiary institutions, especially in colleges of education across the country. Moreover, the study was purely quantitative in nature, which, from my perspective, falls short of providing a subjective view of how the TPB constructs explain students' intentions to study indigenous languages. A qualitative application of TPB is, therefore, needed to provide a better understanding of students' intentions to pursue indigenous languages in school.

## Supporting information

**S1 File.**
(SPV)

## Acknowledgments

I acknowledge the support of Mr. Daniel Baffour-Koduah, Mr. Abdul-Rahman Mutawakil, and Godfred Ntiakoh Bonin. Their support in the administration of the questionnaire facilitated the completion of the entire research.

## Author Contributions

**Conceptualization:** Ernest Nyamekye.

**Formal analysis:** Ernest Nyamekye.

**Investigation:** Ernest Nyamekye.

**Methodology:** Ernest Nyamekye.

**Writing – original draft:** Ernest Nyamekye.

**Writing – review & editing:** Ernest Nyamekye.

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
