## [Decision Letter · Decision Letter 0]

25 Apr 2024

PONE-D-24-07380Indigenous Language Learning in Higher Education in Ghana: Exploring Students’ Behavioural Intentions Using an Extended Theory of Planned BehaviourPLOS ONE

Dear Dr. Nyamekye,

Thank you for submitting your manuscript to PLOS ONE. After careful consideration, we feel that it has merit but does not fully meet PLOS ONE’s publication criteria as it currently stands. Therefore, we invite you to submit a revised version of the manuscript that addresses the points raised during the review process.

We look forward to receiving your revised manuscript.

Kind regards,

Ahmad Samed Al-Adwan

Academic Editor

PLOS ONE

Journal Requirements:

Additional Editor Comments:

Thank you for your submission. There major concerns that need to be addressed as outlined in the review reports.

Reviewers' comments:

Reviewer's Responses to Questions

**Comments to the Author**

1. Is the manuscript technically sound, and do the data support the conclusions?

Reviewer #1: Yes

Reviewer #2: Yes

2. Has the statistical analysis been performed appropriately and rigorously? 

Reviewer #1: Yes

Reviewer #2: Yes

3. Have the authors made all data underlying the findings in their manuscript fully available?

Reviewer #1: No

Reviewer #2: No

4. Is the manuscript presented in an intelligible fashion and written in standard English?

Reviewer #1: Yes

Reviewer #2: Yes

5. Review Comments to the Author

Reviewer #1: Overall, the paper is well-written and holds merit. However, there are several suggestions to enhance the quality of the work:

Introduction Clarity: The main contributions of the paper could be presented more clearly in the introduction. Providing a succinct overview of the key findings and their significance would help orient readers effectively.

Methodology Detail: The methodology section requires more elaboration, particularly concerning the sample frame, sampling technique, and justification of sample size adequacy. These details are crucial for readers to evaluate the robustness and generalizability of the study's findings.

Analytical Approach Justification: A stronger justification behind the use of PLS-SEM as the primary analytical approach is needed. Clarifying whether this choice is based on data distribution characteristics or other methodological considerations would enhance the credibility of the research.

Factor Loading Justification: Consider deleting "att4" from Table 2 based on its factor loading, unless there are other substantive reasons for its inclusion. Ensuring consistency and reliability in factor loadings strengthens the validity of the measurement model.

Common Method Bias Evaluation: The handling of common method bias evaluation needs to be addressed more comprehensively. Incorporating statistical evaluations alongside any methodological approaches employed would enhance the robustness of the findings.

Multi-Collinearity Check: Given the presence of multiple dependent variables, it is essential to perform a multi-collinearity check to ensure the independence of predictor variables. This step would enhance the accuracy and reliability of the regression analyses conducted.

Reviewer #2: I would like to thank for the opportunity to review this manuscript. This manuscript is written on an important and interesting topic, and it reads well. I have only some suggestions to further improve the quality of this manuscript.

Title

The title of this manuscript is an accurate presentation of this manuscript content. However, Authors could simplify and shorten their manuscript title.

Abstract

The abstract provides important information for the reader. However, Authors could add p-values next to beta values.

Main text

Overall, the introduction of this manuscript is well written, but in current forms it is too short. Authors could elaborate the theoretical framework of the theory of planned behaviour. Also, Authors have only briefly mentioned the term motivation. However, recent studies by Hagger and colleagues have put the theory of planned behaviours into larger context, specifically integrated into the trans-contextual model of motivation (TCM. Please see studies by Hagger and colleagues that rely on the TCM. By adding the conceptualization of TCM, one could provide better explanation where language learning intentions come from. Based on the TCM, need-supportive behaviours from significant others are important predictors that ultimately predict intention of different behaviours. For example, Ahmadi and colleagues (2023), have provided very specific list of motivational/need-supportive behaviours that could be highly useful in future studies to better understand and design interventions with the aim to increase language learning intentions.

Ahmadi, A., Noetel, M., Parker, P., Ryan, R. M., Ntoumanis, N., Reeve, J., Beauchamp, M., Dicke, T., Yeung, A., Ahmadi, M., Bartholomew, K., Chiu, T. K. F., Curran, T., Erturan, G., Flunger, B., Frederick, C., Froiland, J. M., González-Cutre, D., Haerens, L., . . . Lonsdale, C. (2023). A classification system for teachers’ motivational behaviors recommended in self-determination theory interventions. Journal of Educational Psychology, 115(8), 1158–1176. https://doi.org/10.1037/edu0000783

Quality of the figures could be better, specifically please see figure 3 and 4.

Please see if there is something missing in table 5.

Discussion of results could be elaborated.

6. PLOS authors have the option to publish the peer review history of their article (what does this mean?). If published, this will include your full peer review and any attached files.

Reviewer #1: **Yes: **Dalin Almbaidin

Reviewer #2: No

---

## [Author Response · Author response to Decision Letter 0]

3 May 2024

Response to Reviewer Comments

I am grateful to the reviewers for their constructive suggestions aimed at enhancing the overall quality of the paper. Below are my responses to their suggestions. Thank You 

Reviewer 1#: 

Introduction

Comment1: Introduction Clarity: The main contributions of the paper could be presented more clearly in the introduction. Providing a succinct overview of the key findings and their significance would help orient readers effectively.

Response: I’m grateful for this comment. However, I believe that the main contribution of a paper, as a general rule of thumb, is presented in the results and discussion section and recapped in the abstract. Presenting the main contribution in the introduction is new to me. I therefore prefer devoting the introduction to the presentation of background issues that frames the importance of this research, as has been done already. However, I am ready to make changes if the journal requirement demands a recap of the main contribution in the introduction. Thank You

Methodology

Comment 1: Methodology Detail: The methodology section requires more elaboration, particularly concerning the sample frame, sampling technique, and justification of sample size adequacy. These details are crucial for readers to evaluate the robustness and generalizability of the study's findings.

Response: I appreciate the reviewer’s comment on this particular issue. As regards the sample frame and sample size, I have already made it clear that the study considered all first-year undergraduate students who have the option to either drop, minor, or major in the Ghanaian language programme as they graduate to second (since this is a general university requirement). In terms of sample size, a total of 180 students sampled out of a general population of 258 is representative and could aid generalizability as supported by Krejcie and Morgan (1970). (see page 11 under the study design and participants). I suppose the reviewer's comment had already been answered in the manuscript. Nonetheless, I would be glad to know if the reviewer has other pertinent issues to address on the sample

Results and Discussions

Comment 1: Analytical Approach Justification: A stronger justification behind the use of PLS-SEM as the primary analytical approach is needed. Clarifying whether this choice is based on data distribution characteristics or other methodological considerations would enhance the credibility of the research.

Response: I have justified using PLS-SEM, indicating that “This statistical approach was considered for the current study because it is usually deemed an appropriate analytical tool when dealing with a complex statistical model involving multiple latent and observed variables (Joe F. Hair et al., 2011; Joseph F. Hair et al., 2013)” (see page 14)

Comment 2: Factor Loading Justification: Consider deleting "att4" from Table 2 based on its factor loading, unless there are other substantive reasons for its inclusion. Ensuring consistency and reliability in factor loadings strengthens the validity of the measurement model.

Response: the factor loading of ‘att4’ was retained in this study because, despite the recommended threshold of .7, a low loading between .4 and .7 could be retained in a model it doesn’t significantly affect other validity and reliability values. As Joseph F. Hair et al. (2021) clearly indicate, low loadings between 0.4 and 0.7 could be retained if their removal does not affect the other reliability and validity issues. 

 Below is a reference that supports the rationale behind the inclusion of att4:

 (Hair, J. F., Hult, G. T. M., Ringle, C. M., Sarstedt, M., Danks, N. P., & Ray, S. (2021). Evaluation of Reflective Measurement Models. In J. F. Hair Jr, G. T. M. Hult, C. M. Ringle, M. Sarstedt, N. P. Danks, & S. Ray (Eds.), Partial Least Squares Structural Equation Modeling (PLS-SEM) Using R: A Workbook (pp. 75-90). Cham: Springer International Publishing.

Comment 3: Common Method Bias Evaluation: The handling of common method bias evaluation needs to be addressed more comprehensively. Incorporating statistical evaluations alongside any methodological approaches employed would enhance the robustness of the findings.

Response: This issue has been resolved. Common method bias has been evaluated and reported in the study (see pages 22). The SPSS output for the analysis is uploaded as a supplementary file

Comment 4: Multi-Collinearity Check: Given the presence of multiple dependent variables, it is essential to perform a multi-collinearity check to ensure the independence of predictor variables. This step would enhance the accuracy and reliability of the regression analyses conducted.

Response: The suggestion was already in the manuscript (See page 21)

Reviewer 2#

Comment 1: Title: The title of this manuscript is an accurate presentation of this manuscript content. However, Authors could simplify and shorten their manuscript title.

Response: I appreciate the reviewers’ suggestions in this regard. However, I believe the title is not too long or difficult to comprehend. If possible, I suggest that the title should remain in its current state. However, would greatly appreciate it if the reviewer suggested a specific alternative

Comment 2: Abstract: The abstract provides important information for the reader. However, Authors could add p-values next to beta values.

Response: As could be observed in the abstract, I have added the p-values for the various statistical relationships in the model. Moreover, a few inconsistencies in the path coefficients have been resolved in the abstract

Comment 3: Main text: Overall, the introduction of this manuscript is well written, but in its current form it is too short. Authors could elaborate the theoretical framework of the theory of planned behaviour. 

Response: I appreciate the reviewers’ submission on the length of the introduction. However, I believe that the introduction is not as short as the reviewer suggests. I say so because from my perspective the relevant issues that grounds the conduct of the study have been pointed out in the introduction. However, I would be grateful and ever ready to include other relevant discourses on the matter the reviewer wishes I should include in the introduction. 

Comment 4: Also, Authors have only briefly mentioned the term motivation. However, recent studies by Hagger and colleagues have put the theory of planned behaviours into larger context, specifically integrated into the trans-contextual model of motivation (TCM. Please see studies by Hagger and colleagues that rely on the TCM. By adding the conceptualization of TCM, one could provide better explanation where language learning intentions come from. Based on the TCM, need-supportive behaviours from significant others are important predictors that ultimately predict intention of different behaviours. For example, Ahmadi and colleagues (2023), have provided a very specific list of motivational/need-supportive behaviours that could be highly useful in future studies to better understand and design interventions with the aim to increase language learning intentions.

Response: I appreciate the recommendation of the reviewer for the suggestion of an alternative theory that could equally be used to provide answers to why students may choose to either pursue or quit pursuing an indigenous language programme. I should be frank to acknowledge that this is my first time coming across the trans-contextual model of motivation (TCM) as a theoretical framework. I therefore think that I can employ this model in future studies. The planning and execution of this paper (including the design of the questionnaire) were based soley on the Theory of Planned Behaviour. Taking the reviewer's suggestion into consideration would call for a restart of the entire data collection process. I thus suggest that in subsequent papers I could consider using the trans-contextual model of motivation (TCM) to provide another perspective on the topic. Thank you. 

Comment 4: The quality of the figures could be better, specifically please see Figures 3 and 4.

Please see if there is something missing in table 5. 

Response: I have maximized the size of both figures to improve the visibility of the figures 

Comment 5: Discussion of results could be elaborated.

Response: I thank the reviewer for this suggestion. From my perspective, the discussion touches on all relevant findings and how they relate to previous literature regarding indigenous language education on Ghana. The reviewer could be more specific on the relevant issues to include to enhance the quality of the discussion. Thank you.

---

## [Editor Report · Decision Letter 1]

13 May 2024

Indigenous Language Learning in Higher Education in Ghana: Exploring Students’ Behavioural Intentions Using an Extended Theory of Planned Behaviour

PONE-D-24-07380R1

Dear Dr. Nyamekye,

We’re pleased to inform you that your manuscript has been judged scientifically suitable for publication and will be formally accepted for publication once it meets all outstanding technical requirements.

Kind regards,

Ahmad Samed Al-Adwan

Academic Editor

PLOS ONE

Additional Editor Comments (optional):

Thank you for submitting the revised version of your paper. The quality of your paper has increased significantly after addressing the reviewers' comments.
---

## [Editor Report · Acceptance letter]

16 May 2024

PONE-D-24-07380R1 

PLOS ONE

Dear Dr. Nyamekye, 

I'm pleased to inform you that your manuscript has been deemed suitable for publication in PLOS ONE. Congratulations! Your manuscript is now being handed over to our production team.

Kind regards, 

on behalf of

Professor Ahmad Samed Al-Adwan 

Academic Editor

PLOS ONE